# Comparative Study on the Structural Properties and Bioactivities of Three Different Molecular Weights of *Lycium barbarum* Polysaccharides

**DOI:** 10.3390/molecules28020701

**Published:** 2023-01-10

**Authors:** Wenjun Zeng, Lulu Chen, Zhihui Xiao, Yanping Li, Jianlong Ma, Jianbao Ding, Jin Yang

**Affiliations:** 1School of Chemistry and Chemical Engineering, North Minzu University, Yinchuan 750021, China; 2Key Laboratory for Chemical Engineering and Technology, North Minzu University, State Ethnic Affairs Commission, Yinchuan 750021, China; 3South China Sea Institute of Oceanology, Chinese Academy of Sciences, Guangzhou 510301, China; 4Ningxia Wuxing Science and Technology Co., Ltd., Yinchuan 750021, China; 5Ningxia Research Center for Natural Medicine Engineering and Technology, Yinchuan 750021, China; 6College of Chemistry and Chemical Engineering, Ningxia University, Yinchuan 750021, China

**Keywords:** *Lycium barbarum* polysaccharides, molecular weight, chemical characteristics, antioxidant activity, immunomodulatory activity

## Abstract

The molecular weight, the triple-helix conformation, the monosaccharide content, the manner of glycosidic linkages, and the polysaccharide conjugates of polysaccharides all affect bioactivity. The purpose of this study was to determine how different molecular weights affected the bioactivity of the *Lycium barbarum* polysaccharides (LBPs). By ethanol-graded precipitation and ultrafiltration membrane separation, one oligosaccharide (LBPs-1, 1.912 kDa) and two polysaccharides (LBPs-2, 7.481 kDa; LBPs-3, 46.239 kDa) were obtained from *Lycium barbarum*. While the major component of LBPs-1 and LBPs-2 was glucose, the main constituents of LBPs-3 were arabinose, galactose, and glucose. LBPs-2 and LBPs-3 exhibited triple-helix conformations, as evidenced by the Congo red experiment and AFM data. Sugar residues of LBPs-2 and LBPs-3 were elucidated by NMR spectra. The polysaccharides (LBPs-2 and LBPs-3) exhibited much higher antioxidant capacities than oligosaccharide (LBPs-1). LBPs-3 showed higher oxygen radical absorbance capacity (ORAC) and superoxide dismutase (SOD) activity than LBPs-2, but a lower capability for scavenging ABTS^+^ radicals. In zebrafish, LBPs-2 and LBPs-3 boosted the growth of T-lymphocytes and macrophages, enhanced the immunological response, and mitigated the immune damage generated by VTI. In addition to the molecular weight, the results indicated that the biological activities would be the consequence of various aspects, such as the monosaccharide composition ratio, the chemical composition, and the chemical reaction mechanism.

## 1. Introduction

Polysaccharides are chain-like polymers of biological origin (including animals, plants, and fungi), usually composed of more than 10 monosaccharides joined by α- and/or β-glycosidic linkages by dehydration [1]. Numerous studies have shown that polysaccharides have a variety of bioactivities, including anticancer [2], antioxidant [3], anti-aging [4], immunomodulation [5,6], and so on. They are also utilized as drug carriers [7]. In recent years, polysaccharides have garnered a great deal of interest due to their rising prevalence in functional foods, pharmaceuticals, cosmetics, etc. [1].

The bioactivities of polysaccharides are dependent on their molecular weight [8,9,10], the triple-helix conformation [11], the monosaccharide content, the glycosidic bonds mode [10], and the polysaccharide conjugates [12]. *Lycium barbarum* polysaccharides (LBPs) are water-soluble glycoconjugates isolated from Goji berry or wolfberry (Solanaceae), one of the most popular traditional medicine materials. Phytochemical studies have revealed that LBPs are the most important group of substances in this herbal medicine, responsible for the majority of the pharmacological properties, including antioxidant, immunomodulatory, anticancer, and neuroprotective actions [13,14,15]. Feng et al. investigated the immunomodulatory effects of LBPs with varying molecular weights [16], but information on the structural properties is scant. Due to the complexity and diversity of their structures, it is challenging to determine which structural characteristics of LBPs may influence their biological activity. It would be beneficial to understand the molecular mechanism of these macromolecules to characterize their structural characteristics as thoroughly as possible while describing their biological effects.

Using response surface methodology (RSM), the influence of several extraction parameters (extraction temperature, duration, and water-to-raw material ratio) on the yield of LBPs was evaluated in this work. One oligosaccharide (LBPs-1), and two polysaccharides (LBPs-2 and LBPs-3) were separated from LBPs by graded ethanol precipitation and ultrafiltration membrane separation. The polysaccharide fractions were then characterized using high-performance size exclusion chromatography–refractory index detector–multi-angle laser light scattering (HPSEC-RID-MALLS), high-performance anion exchange chromatography with pulsed amperometric detection (HPAEC-PAD), Fourier transform infrared spectroscopy (FTIR), atomic force microscopy (AFM), and nuclear magnetic resonance (NMR). Furthermore, the in vitro antioxidant activities of three polysaccharides with different molecular weights, including the oxygen radical absorbance capacity (ORAC), the ABTS^+^ radical scavenging capacity, and the superoxide dismutase (SOD) activity, were evaluated. Finally, the immunoregulatory activity of LBPs-2 and LBPs-3 in the T-lymphocyte and macrophage damage models caused by vinorelbine tartrate injection (VTI) in zebrafish were examined.

## 2. Results and Discussion

### 2.1. Optimization of the Crude Polysaccharide Extraction Process by RSM

#### 2.1.1. Single-Factor Tests

There are presently no repeatable preparation methods to guarantee the uniformity of samples for activity tests. The nature of LBPs can vary depending on the extraction procedure, resulting in the variation of bioactivities [10,17]. Various methods, such as hot water extraction, ultrasonic-assisted extraction, microwave-assisted extraction, etc., can be used to extract polysaccharides, with hot water extraction being the most common method for the preparation of polysaccharides [10]. However, the yield of this method is limited and the process is time-consuming [1]. To improve the extraction rate, RSM is usually used to optimize the extraction process parameters of polysaccharides [18]. RSM can reduce the number of experiments required to evaluate multiple parameters and their interactions and generate a mathematical model to determine the optimal value [19]; this process is relatively time- and labor-efficient. During the hot water extraction processes, the extraction temperature, the time, and the water-to-wolfberry ratio significantly impacted the yield of polysaccharides [18,20]. Therefore, these variables were selected for the single-factor tests.

Under constant conditions (the extraction time was 60 min and the water-to-wolfberry ratio was 12 mL/g), the effects of the temperature on the extraction yield of LBPs were tested twice. As shown in Figure 1A, the extraction yield of LBPs increased rapidly from 60 °C to 90 °C, reaching 13.59%. This pattern was comparable to the findings of previous studies on polysaccharide extraction [18,21]; the extraction yield improved somewhat as the temperature was increased. Even though the polysaccharide extraction yield was highest at 100 °C, increasing the temperature would raise the cost of the extraction process from an industrial perspective. Therefore, 90 °C was chosen as the center point of the temperature during the response surface tuning.

Under fixed conditions (the extraction temperature was 90 °C and the water-to-raw material ratio was 12 mL/g), the effects of time on the extraction yield of LBPs were studied twice. As depicted in Figure 1B, as the extraction period increased from 30 to 60 min, the extraction yield of LBPs initially grew and then tended to plateau; nevertheless, there was no statistically significant difference (*p* > 0.05). Thus, 60 min was selected as the experiment’s central time frame.

Under controlled conditions (the extraction temperature was 90 °C and the duration was 60 min), the effects of the water-to-raw material ratio on the extraction yield of LBPs were investigated twice. As shown in Figure 1C, the extraction yields increased as the water-to-raw material ratios increased between 9 and 15, peaking at 13.93% at 15 mL/g, and subsequently fell as the water-to-wolfberry ratio continued to grow. This phenomenon may be caused by the high water-to-raw material ratio, which results in a substantial concentration difference between the internal and external solvents of plant cells. However, a higher water-to-raw material ratio may reduce the molecular interactions [22]. Consequently, 15 mL/g was chosen as the central water-to-raw material ratio for the response surface experiment.

#### 2.1.2. Response Surface Analysis

As mentioned in Table 1, 17 trials were conducted to optimize the 3 individual extraction parameters for the BBD experiment. The yield of LBPs ranged from 10.91% to 15.24%, with the highest yield obtained under the following experimental conditions: 100 °C extraction temperature, 75 min extraction duration, and the 15 mL/g water-to-raw material ratio. Through multiple regression analysis of experimental data, a correlation between the response value and test variables was established as follows:Y = 13.95 + 1.04A + 1.33B + 0.30C + 0.58AB − 0.094AC + 0.15BC − 0.44A^2^ − 1.02B^2^ − 0.27C^2^
where Y is the LBPs yield (%), and A, B, and C are the extraction temperature, the time, and the water-to-raw material ratio, respectively.

Typically, F and *p* are values employed to assess the significance of the model coefficients. In general, the model and its variables are more significant when F values are relatively high and *p* values are low. According to Table 2, the model’s F value was 24.28 and its *p* value was 0.0002, suggesting that it was extremely significant. The lack-of-fit F value and *p* value were 5.94 and 0.0590, respectively, while the model’s correlation coefficient R^2^ was 0.969, indicating a high degree of fitting and a small error. The regression equation adequately described the relationship between each factor and the response value. Moreover, the *p* values of the coefficients A, B, AB, A^2^, and B^2^ were less than 0.05, implying that they had significant effects on the extraction yield of polysaccharides, whereas the *p* values of the coefficients C, AC, BC, and C^2^ were greater than 0.05, revealing that they had no significant effect on the extraction yield of polysaccharides.

Design-Expert (Version 10) was used to produce the 3D response surface and contour map, as depicted in Figure 2, to predict the relationship between the dependent and independent variables. These diagrams illustrate the interaction between the two variables as well as the relationship between the response and the experimental level for each variable. The steepness of the 3D graph indicates the degree of influence between the variables on the value of the response. Figure 2A,D display the influence of the extraction temperature (A) and the time (B) on the extraction yield of LBPs when the water-to-raw material ratio (C) was held constant. The steep response surface shown in Figure 2D suggests a considerable interaction between the extraction temperature and the time. However, Figure 2E’s response surface was nearly flat, revealing that the interaction between the extraction temperature and the water-to-raw material ratio was not significant. Similar tendencies were observed for the extraction period and the water-to-raw material ratio (Figure 2C,F). Based on the preceding RSM analysis, it was hypothesized that the extraction temperature and the duration significantly affected the extraction yield of LBPs.

The RSM analysis provided the optimal conditions for applying the model equation: the extraction temperature was 100 °C, the time was 74.82 min, and the water-to-material ratio was 16.85 mL/g. Under the optimum terms, the maximum yield of LBPs was 15.57%. In the actual operation, these were the ideal conditions: the extraction temperature was 100 °C, the extraction period was 75 min, and the water-to-material ratio was 17 mL/g. Five validation experiments were conducted under the changed optimal conditions to confirm that the predicted result did not differ from the experimental value. The results indicated that the actual extraction yield was 15.58 ± 0.19%, which was nearly identical to the expected yield of 15.57%. The outcomes demonstrated that the experimental model was reliable and appropriate for predicting the extraction conditions of LBPs.

### 2.2. The Physicochemical Properties of LBPs with Different Molecular Weights

Recent studies have suggested that polysaccharides with different molecular weights exhibit different biological activities [16,23]. Hence, to facilitate the efficient development of functional polysaccharides, it is necessary to investigate the physicochemical properties and structures of the different molecular weight polysaccharides separated by this method and to understand the relationships between their structure and bioactivities [24].

Ultrafiltration membrane separation technology based on membrane separation methods is appropriate for polysaccharide enrichment and purification because of its simple operation, absence of chemical ingredients, short cycle, and high throughput [25]. Using ultrafiltration membrane separation, two polysaccharides (LBPs-2 and LBPs-3) and one oligosaccharide (LBPs-1) were separated from LBPs. The percentages of LBPs-1, LBPs-2, and LBPs-3 were about 42.62%, 3.41%, and 10.22% in the crude LBPs, respectively. As shown by the chromatogram in Figure 3A, LBPs-1, LBPs-2, and LBPs-3 had absolute molecular weights and polydispersity indices of 1.912 kDa and 1.514, 7.481 kDa and 1.324, and 46.239 kDa and 1.192, respectively.

Previous research has indicated that the biological activity of polysaccharides was dependent on their chemical composition [7,26,27]. We analyzed the total saccharide, protein, and polyphenol contents of LBPs with various molecular weights. According to Table 3, there was no significant difference in the total sugar content of the LBPs with different molecular weights (*p* > 0.05), but there were significant differences in the total protein content and total polyphenol content (*p* < 0.05). LBPs-3 had the highest protein content, while LBPs-2 contained the most polyphenol overall. Moreover, in our previous study, it was shown that these polyphenols were present in the bound form with polysaccharides, i.e., polysaccharide–polyphenolic conjugates, and it has been suggested that polysaccharide–polyphenolic conjugates exhibited better biological activity [28,29]. Therefore, these differences in chemical composition may have impact their biological activity.

The specific structure and bioactivity of polysaccharides are determined by their monosaccharide building blocks. The monosaccharide compositions of LBPs were measured by the HPAEC-PAD method, and the findings are presented in Table 3 and Figure 3B. LBPs-2 and LBPs-3 share the same monosaccharide composition; however, LBPs-1 lack fucose. In LBPs-1 and LBPs-2, glucose dominated the monosaccharides, while arabinose, galactose, and glucose were prevalent in LBPs-3. Compared to LBPs-1 and LBPs-2, LBPs-3 had much less glucose and significantly more rhamnose, arabinose, and galactose. The results suggested that, in this case, the proportion of glucose was dominant in polysaccharides with low Mw, and replaced by arabinose and galactose in high Mw LBPs.

### 2.3. The Structural Features and Conformation of LBPs with Different Molecular Weights

Through FT-IR spectroscopy, the structures and types of polysaccharide functional groups were investigated. Figure 4 illustrates the infrared characteristic curves of the three LBPs samples. The intense absorption peak at 3369 cm^−1^ represented the O-H and N-H stretching vibration, characteristic of polysaccharides [30]. The absorption peak at 2936 cm^−1^ corresponded to the tensile vibration of C-H [31]. The relatively strong absorption peak at around 1612 cm^−1^ indicated the presence of C=O [32]; the existence of C=O implied that the polysaccharides contained an amide bond or carboxylic acid, suggesting that the LBPs were protein-binding polysaccharides. The band at 1421 cm^−1^ could be attributed to the absorption of the N-H groups from the amide bond (-CONH) of the polysaccharides. In the region between 1257 cm^−1^ and 1055 cm^−1^, the presence of C-O-C and C-O-H link bonds was observed, and combined the absorption peaks at 820 cm^−1^ and 777 cm^−1^ could establish the presence of pyranose [33]. Moreover, the distinctive peaks at 865 cm^−1^ and 920 cm^−1^ denoted the α- and β-glycosidic bonds [34,35]. These results suggested that the three LBPs with different molecular weights had no detectable effects on the arrangement of the glycosidic linkages or main functional groups.

The Congo red experiment detected whether the polysaccharides had triple-helix conformation. Figure 5A,B depict the λ_max_ and λ_max_ − λ_Blank_ values of the complexes formed between Congo red and LBPs in NaOH solutions of different concentrations. In 0–0.3 M NaOH solutions, LBPs-2 and LBPs-3 exhibited distinct redshifts relative to the Congo red and the LBPs-1-Congo red complex, indicating that they possessed a triple-helix conformation at a low alkaline concentration [36,37,38]. The observation suggested that the triple-helix conformations of LBPs may exist only in substances with a molecular weight greater than 3 kDa. In addition, the maximum displacement of LBPs-3 in the 0.025 M NaOH solution was 14 nm, which was more obvious than that of LBPs-2, indicating that the molecular weight had a significant influence on its conformation. This phenomenon may be caused by the increased intra- and intermolecular hydrogen bonds in polysaccharides that have a high molecular weight, which is more favorable to the formation of the triple-helix conformation.

Using AFM, the triple-helix conformations of LBPs were observed more directly. Figure 6A–D show planar and 3D AFM images of the two polysaccharides with triple-helix conformations in the Congo red experiment. Indirectly reflecting changes in molecular structure, alterations in height can be used to aid in the investigation of the triple-helix conformations. Adamcik [39] and Xiao [40] et al. demonstrated that double- and triple-helical aggregates were almost two and three times as tall as the single chain, respectively. The heights of the single, double, and triple chains in LBPs-2 and LBPs-3 were 1.0 ± 0.1, 2.0 ± 0.2, and 3.0 ± 0.1 nm, respectively. These results suggest the LBPs-2 and LBPs-3 possess the triple-helix conformation. Divergent perspectives exist over whether the stable region (0.05–0.19 mol/L NaOH) of Congo red’s absorption curve is required for determining the triple-helix conformation of polysaccharides [11]. The AFM images of LBPs-2 and LBPs-3 revealed that this conformation may be recognized as long as a significant redshift was exhibited in the low-alkali concentrations.

The ^1^H NMR, ^13^C NMR, HSQC, HMBC, and ^1^H-^1^H COSY data on LBPs-2 and LBPs-3 were collected by NMR to comprehend the structural detail of the polysaccharides (Appendix A). Based on the NMR spectra of LBPs-2 and LBPs-3 and the pertinent literature [34], the following inferences were made. The ^1^H NMR (D_2_O, 700 MHz) signals of LBPs-2 and LBPs-3 were mainly concentrated within the range of 3.2–5.5 ppm. As suggested by ^13^C NMR signal analysis (D_2_O, 175 MHz), the signals of LBPs-2 and LBPs-3 were mainly observed in the range between 60 and 120 ppm. Moreover, to understand the glycosidic bond types and connection modes of LBPs-2 and LBPs-3, we speculated them by combining HSQC, HMBC, and ^1^H-^1^H COSY; the results are listed in Table 4 and Table 5. They all had seven glycosyl residues signal. LBPs-2 included →4)-β-Gal*p*-(1→, α-Glc*p*-(1→, →3)-α-Glc*p*-(1→, β-Glc*p*-(1→, →3,4)-β-Ara*p*-(1→, →3)-α-Ara*p*-(1→ and →4)-α-D-Glc*p*A-(1→, while LBPs-3 consisted of →3,4)-α-Ara*p*-(1→, →3,4)-α-Gal*p*-(1→, →3)-α-Gal*p*-(1→, →4)-β-Ara*p*-(1→, β-Ara*p*-(1→, →3,4)-α-Gal*p*-(1→ and →3)-α-Glc*p*A-(1→.

### 2.4. The Antioxidant Activities of LBPs In Vitro

In vitro, antioxidant activity is an essential measure of the bioactivity of natural polysaccharides. To investigate the antioxidant activity of LBPs with different molecular weights, the ABTS^+^ radical scavenging ability, the oxygen radical absorbance capacity (ORAC), and the superoxide dismutase (SOD) activity were measured. ABTS^+^ and ORAC studies were performed to examine the ability of the samples to scavenge free radicals, although the reaction processes involved were distinct. The former was based on a single electron transfer (SET) reaction pathway to scavenge free radicals [41], while the latter involved hydrogen atom transfer (HAT) [42]. The SOD assay evaluated the activity of superoxide dismutase, an important antioxidant enzyme that scavenges superoxide anion radicals. In addition, SOD can enhance immunity and improve resistance to diseases caused by free radicals [43], which may also serve as the initial step in preventing oxidative damage [44].

As shown in Figure 7, all the LBPs exhibited antioxidant activity in vitro. In the ORAC and SOD experiments, LBPs-3 displayed the highest activity (*p* < 0.01), with ORAC and SOD values of 1118.05 ± 45.94 μmoL Vc/g DW, 693.79 ± 34.41 U/DW, which were 2.04 and 1.31, and 4.53 and 1.84 times greater than those of LBPs-1 and LBPs-2, respectively. Despite this, the ABTS data demonstrated that LBPs-3 was not the most active. LBPs-2 had a TEAC value of 28.99 μmoL/g DW, which was 2.32 and 1.12 times higher than LBPs-1 and LBPs-3 (*p* < 0.01), respectively. The antioxidant potential of the polysaccharides was related to their chemical makeup, with polyphenols and polysaccharide-protein residues capable of contributing H-atoms playing a crucial role in the HAT-based tests [45]. In light of their increased protein and polyphenols, the ORAC value and SOD activity of LBPs-2 and LBPs-3 were higher than those of LBPs-1. However, based on the SET mechanism [46,47], the polyphenols were the primary factors that influenced antioxidant effects, resulting in an increased TEAC value for LBPs-2. Moreover, the proportions of rhamnose, arabinose, and galactose in the composition of the monosaccharides were also considered to be associated with antioxidant activity [12,48]. Zhang et al. [49] discovered that rhamnose played an important role in radical-scavenging capacities, while Kang et al. [50] noted that an increase in the ratio of rhamnose, arabinose, and galactose boosted antioxidant activity. In addition, for different types of polysaccharides, the molecular weight is a significant influence in determining the antioxidant capacity [51]. In several prior investigations, polysaccharides with greater molecular weights showed rather strong antioxidant activity [52,53]. In contrast, Xu [54] and Liu [55] et al. found that low-molecular-weight polysaccharides held higher antioxidant activity than high-molecular-weight polysaccharides. According to our findings, the varied antioxidant activity of LBPs with the varied molecular weights did not contribute to a single cause. Several interrelated factors, such as the molecular weight, the monosaccharide composition ratio, the chemical composition, and the chemical reaction mechanism [54,56,57], lead to the biological activities.

### 2.5. In Vivo Immunomodulatory Effect of LBPs-2 and LBPs-3

#### 2.5.1. MTC Assessment

Due to their advantages of repeatability and high-throughput behavioral screening, zebrafish have emerged as a model for toxicology and pharmacology [58]. The toxicity of the polysaccharides was evaluated by measuring the larvae survival rate (Figure 8). In the dose range of 125–500 μg/mL, there was no significant change in the survival rate of zebrafish incubated with LBPs, indicating no substantial toxicity to the experimental animals. When the concentration reached 1000 μg/mL, the transgenic macrophage fluorescent zebrafish was considerably influenced. Therefore, 125, 250, and 500 μg/mL were selected for the following tests.

#### 2.5.2. The Effect of LBPs-2 and LBPs-3 on T-Lymphocytes and Macrophages in VTI-Treated Zebrafish

Polysaccharides with immunostimulatory features can interact directly or indirectly with the immune system by initiating cellular and molecular events that contribute to the immune system activation [59]. T-lymphocytes and macrophages are the primary cells targeted by these events. The former is associated with cellular immune responses to intracellular pathogens [60], while the latter is an important component of the host’s immune defense system, allowing macrophages to collaborate with other cell types (such as T-lymphocytes and neutrophils) to defend against external damage [61,62,63,64]. Polysaccharides exert immunomodulatory effects on these two types of cells through the generation of reactive oxygen species (ROS), cell proliferation, and secretion of cytokines, among other mechanisms [63]. T-lymphocytes and macrophages were fluorescently labeled with red and green fluorescence, respectively, to determine whether exposure to LBPs-2 and LBPs-3 improved the immune system abnormalities produced by VTI in zebrafish [65].

As shown in Figure 9, the fluorescence intensity of both the T-lymphocytes and the macrophages in the model group treated with VTI was significantly reduced (*p* < 0.01), by 59.88% and 29.76%, respectively, when compared to the control group, indicating that VTI diminished zebrafish immunity. Treatment with different concentrations of LBPs-2 and LBPs-3 increased the fluorescence intensity of both cell types in the range of 125–500 μg/mL in a dose-dependent manner. At higher doses (250–500 μg/mL), the value-added effect of the two polysaccharides on T-lymphocytes and macrophages did not differ significantly (*p* > 0.05). At a concentration of 125 μg/mL, however, the value-added effect of LBPs-3 on T-lymphocytes was superior to that of LBPs-2 (*p* < 0.01). Feng et al. discovered that the immunoreactivity of the polysaccharide fraction > 10 kDa (27.7 kDa) in LBP was better than that of the <10 kDa fraction (6.99 kDa), suggesting that molecular weight was an essential factor affecting their immunoreactivity. In addition, it has been demonstrated that the arabinogalactan backbone is present in many LBPs with immunomodulatory action; hence, polysaccharides with relatively high levels of arabinose and galactose have stronger immunomodulatory activity [66,67,68]. Consequently, the different observations at a concentration of 125 μg/mL may be attributable to the different physicochemical properties and structures of the two polysaccharides [69].

## 3. Materials and Methods

### 3.1. Materials and Reagents

Goji berries were purchased from Zhongning International Wolfberry Trading Center, Zhongning County, Ningxia. The samples were identified by Prof. J. Ding of Ningxia Wuxing Science and Technology Co., Ltd. (Yinchuan, China), and the voucher specimens were stored at North Minzu University. The Goji berries were kept at −20 °C before use. The monosaccharide standards, Congo red, L-ascorbic acid, fetal bovine serum (FBS), and DMEM reagents were purchased from Titan Scientific Co., Ltd. (Shanghai, China). All other chemicals and solvents utilized in this investigation were of analytical quality.

### 3.2. Extraction of Crude Polysaccharides

Firstly, the Goji berries were extracted twice with purified water and the solution was collected by centrifugation (6000 rpm, 5 min). All supernatants were concentrated to the appropriate volume. Then, 95% ethanol (*v*/*v*) was added until the final ethanol concentration reached 50%, and the mixture was maintained overnight. The precipitate was removed using centrifugation. Then, 95% EtOH (*v*/*v*) was again added to the solution to adjust the alcohol concentration to 85% and the mixture was stored for 12 h [70]. Centrifugation was used to collect the precipitate. The retentate was dissolved in water and further deproteinated with Sevag method [71]. The aqueous phase was concentrated and lyophilized (FD-1C-50, Beijing Biocool Co., Ltd., Beijing, China) to obtain crude LBPs.

The percentage polysaccharide yield (%) was computed as follows:$$\mathrm{Polysaccharide}\;\mathrm{yield}\;\left(\%\right)=\left(\frac{\mathrm{weight}\;\mathrm{of}\;\mathrm{dried}\;\mathrm{crude}\;\mathrm{polysaccharides}}{\mathrm{weight}\;\mathrm{of}\;\mathrm{wolfberry}}\right)\times 100\%$$

Numerous studies have demonstrated that three single factors (extraction temperature, extraction duration, and water-to-raw material ratio) are the most influential on the yield of the hot water extraction technique [18,20]. Consequently, these variables were selected for the single-factor tests. The extraction temperature was set between 60 and 100 °C, the extraction period was set from 30 to 150 min, and the ratio of the solvent to material ranged from 9:1 to 21:1. During the optimization of experimental variables, just one factor in each experiment was changed, while the remaining factors remained unchanged.

Three single factors (the extraction temperature, the extraction duration, and the water-to-raw material ratio) were optimized using Box–Behnken (BBD)-RSM, and three-factor and three-level response surface analyses were performed using the extraction yield of the polysaccharides as the response value. The three variables were labeled A, B, and C and separated into three tiers. The relative codes for the low, medium, and high values are −1, 0, and +1 (Table 1). The entire design comprised 17 experimental locations chosen at random.

### 3.3. Separation of Polysaccharides

We extracted the LBPs under the optimized conditions, followed by ultrafiltration membrane separation technology to yield polysaccharides with different molecular weights [72]. The crude polysaccharides (10 g) were dissolved in 60 mL of distilled water. After thoroughly dissolving the crude polysaccharides, the monosaccharides and some oligosaccharides were removed using a 1 kDa ultrafiltration tube (Pall Corp., NewYork, NY, USA). The polysaccharide fractions over 1 kDa were then separated again by ultrafiltration tubes of 3 k and 10 kDa (Merck Millipore Co., Ltd., Darmstadt, Germany), and the liquid from each fraction was collected and lyophilized to obtain LBPs-1 (1–3 kDa), LBPs-2 (3–10 kDa), and LBPs-3 (>10 kDa), respectively.

### 3.4. Molecular Weight Analysis

As previously described by Hu et al. [73], the molecular weight was measured by the HPSEC-RID-MALLS system. The system was equipped with a liquid phase (U3000, Thermo Fisher Scientific, Waltham, MA, USA), a differential detector (OPTILAB T-REX, Wyatt Technology Corp., Santa Barbara, CA, USA), a laser light scattering detector (DAWN HELIOS Ⅱ, Wyatt Technology Corp., Santa Barbara, CA, USA), and an Ohpak SB series gel exclusion column (300 × 8 mm, Shodex Co., Ltd., Tokyo, Japan).

The samples were dissolved in 0.1 M NaNO_3_ aqueous solution (containing 0.02% NaN_3_, *w*/*w*) to a final concentration of 1 mg/mL and filtered through a 0.45 μm microporous filtering film (Pall Corp., New York, NY, USA). Then, 0.02% NaN_3_ aqueous solution treated by a 0.45 μm filter membrane was used as the mobile phase. The flow rate was 0.40 mL/min, and the injection volume was 100 μL. Data acquisition and analysis were performed by Astra software (version 6.1).

### 3.5. Determination of the Chemical Composition and Monosaccharide Composition

Using the phenol-sulfuric acid method with glucose as the standard, the total saccharide content was calculated [26]. The protein concentration was measured by a BCA protein assay kit (P0012S, Beyotime Biotechnology Corp., Shanghai, China), using BSA as the standard [74]. The total polyphenol content was determined by the Folin–Ciocalteau method [75] with gallic acid as a standard.

The composition of the monosaccharides was assessed via the HPAEC-PAD method [76]. The samples (5 mg) were hydrolyzed for 2 h at 121 °C with 1 mL of 2 M trifluoroacetic acid (TFA) in a sealed tube. At 50 °C, the residual trifluoroacetic acid was removed by a rotary evaporator. The residue was redissolved in methanol (4 mL) and dried five times, after which it was dissolved in distilled water and injected into the HPAEC-PAD, which consisted of an ICS-5000 system (Thermo Fisher Scientific, Waltham, MA, USA) equipped with a Dionex™ CarboPac™ PA20 (150 mm × 3 mm, 10 μm, Dionex Corp., Sunnyvale, CA, USA) and a pulsed amperometric detector.

### 3.6. FT-IR Spectroscopy Analysis

Using the KBr pellet method, the Fourier transform infrared (FT-IR) spectra of LBPs were recorded by a FTIR spectrometer (FTIR650, Tianjin Gangdong Sci. & Tech. Co., Ltd., Tianjin, China) in the scanning range from 4000 to 400 cm^−1^ with a resolution of 4 cm^−1^. All spectra were scanned at least 5 times.

### 3.7. Congo Red Experiment

The Congo red experiment studied the triple-helix conformation of the samples [34]. Briefly, 2 mg/mL of the LBPs solutions was combined with 160 μg/mL of the Congo red solution and the NaOH solutions with concentrations of 0 mol/L, 0.1 mol/L, 0.2 mol/L, 0.4 mol/L, 0.8 mol/L, 1.2 mol/L, 1.6 mol/L, and 2.0 mol/L, respectively, and the blank group consisted of distilled water. The absorption wavelength of the Congo red in varying concentrations of the NaOH solution was measured using an ultraviolet spectrophotometer (TU-1901, Beijing General Analytical Instrument Co., Ltd., Beijing, China) scanning between 400 and 600 nm.

### 3.8. Atomic Force Microscopy (AFM) Analysis

The stereoscopic structure of the polysaccharides was observed by AFM (Brucker Icon, Bruker Corp., Karlsruhe, Germany). The polysaccharides were dissolved in a 10 μg/mL solution with ultrapure water. After continuous ultrasound for 10 min, 2 μL of the sample solutions was deposited on a mica carrier, dried overnight at room temperature, and then photographed for analysis under a microscope.

### 3.9. Nuclear Magnetic Resonance (NMR) Analysis

The polysaccharides (40 mg) were dissolved in 10 mL of D_2_O, mixed sufficiently, and transferred to the NMR tube using a syringe. ^1^H, ^13^C, HSQC, HMBC, and ^1^H-^1^H COSY NMR were performed on an Advance III HD 700 NMR spectrometer (Bruker Corp., Karlsruhe, Germany) equipped with a cryogenic probe at room temperature.

### 3.10. Determination of the In Vitro Antioxidant Activities of LBPs

#### 3.10.1. Oxygen Radical Absorbance Capacity (ORAC) Assay

The ORAC assay was conducted using modifications to the method described by Huang et al. [77]. The samples were diluted in 75 mM phosphate buffer of (pH 7.4). The experiment was performed on black 96-well plates (Titan Scientific Co., Ltd., Shanghai, China). Each well contained 20 µL of a sample or the L-ascorbic acid standard solution, together with 200 µL of fluorescein solution (to achieve a final concentration of 0.96 µM). The blank consisted of 75 mM phosphate buffer (pH 7.4). The plates were incubated for 20 min at 37 °C on a multi-function enzyme labeling device (Varioskan LUX, Thermo Fisher Scientific Co., Ltd., Shanghai, China). Following incubation, 20 µL of 119.4 mM AAPH was added to each well. The fluorescence conditions were as follows: an excitation wavelength of 485 nm, an emission wavelength of 535 nm, and 71 cycles with measurement at 2 min intervals. The ORAC results were measured in terms of the L-ascorbic acid equivalents per gram of dry weight (DW).

#### 3.10.2. Scavenging Experiment of LBPs on ABTS^+^

The ABTS radical scavenging experiment was determined using a total antioxidant capacity assay kit with a rapid ABTS method (S0121, Beyotime Biotechnology Corp., Shanghai, China). The experiment was performed in 96-well plates in which the samples were diluted with PBS solution. Each well contained 10 µL of the sample or Trolox standard, 20 µL peroxidase, and 170 µL ABTS working solution. As a control, PBS or double steaming water was employed. After mixing, the samples were incubated at room temperature for 6 min; then, the absorbance was measured at 414 nm using the multi-function enzyme labeling equipment. The total antioxidant capacity was indicated by the Trolox-equivalent antioxidant capacity (TEAC).

#### 3.10.3. Superoxide Dismutase (SOD) Activity

The SOD activity was assayed using a superoxide dismutase activity test kit (BC0175, Beijing Solarbio Science & Technology Co., Ltd., Beijing, China). Various solutions were added by following the kit instructions. The sample was then agitated for 10 s within the multi-function enzyme labeling equipment and incubated at 37 °C for 30 min. The absorbance was measured at 560 nm, and the SOD activity was estimated using the dry weight and represented as units per gram of DW. In the xanthine oxidase-coupled reaction system described previously, the inhibition rate was regulated between 30 and 70%, and the SOD activity in the reaction system was defined as the unit of enzyme activity.

### 3.11. In Vivo Immunoregulatory Activity

#### 3.11.1. Zebrafish Husbandry and Egg Collection

Transgenic T-lymphocyte fluorescent zebrafish and transgenic macrophage fluorescent zebrafish (experimental animal use permit number: SYXK 2012-0171) were provided by Hunter Biotechnology Co., Ltd. (Hangzhou, China). According to the standard zebrafish breeding protocol [78], the zebrafish were maintained in a controlled environment at 28 °C with a 14:10 h day/night cycle and were fed brine shrimp twice daily in water with a conductivity of 450–550 μS/cm and a pH range of 6.5 to 8.5. The experimental animals were bred by natural mating. Four days post fertilization (dpf) transgenic T-lymphocyte fluorescent zebrafish and three days post fertilization (dpf) transgenic macrophage fluorescent zebrafish were utilized to evaluate the minimum toxic concentration (MTC) and the immunoregulatory action of LBPs, respectively. All studies were performed in a room with a temperature of 28 °C. All animal treatments were authorized by North Minzu University’s Ethical Review Committee and Laboratory Animal Welfare Committee (Approval No. 2022-9).

#### 3.11.2. MTC Assessment of LBPs-2 and LBPs-3 Model Zebrafish

Random selection of transgenic zebrafish and intravenous administration of 0.2 mg/mL vinorelbine tartrate injection (VTI, Jiangsu Hansoh Pharma Co., Ltd., Hangzhou, China, lot number 140501) were used to construct zebrafish models of immune vulnerability [79]. Randomly, the model zebrafish were transferred to 6-well plates containing 3 mL of LBPs (at concentrations of 0, 125, 250, 500, 1000, and 2000 μg/mL) in medium in each well. In total, 30 fish per well were incubated in an incubator at 28 °C for 24 h. Simultaneously, a normal control group and a model control group were monitored. In each experimental group, the morality and toxicity of zebrafish were tallied, and the MTC values were calculated.

#### 3.11.3. Effect of LBPs-2 and LBPs-3 on the T-Lymphocytes and Macrophages in VTI-Treated Zebrafish

Randomly dividing the immunosuppressed zebrafish into experimental, normal control, and model control groups, 30 fish per well were transferred to 6-well plates. The experimental groups of zebrafish were administered LBP aqueous solutions at doses of 125, 250, and 500 μg/mL, respectively. After incubation for 24 h at 28 °C, 10 fish from the experimental groups were randomly selected and photographed using a fluorescence microscope (AZ100, Nikon Corp., Tokyo, Japan). The T-lymphocyte and macrophage fluorescence intensity of the model zebrafish was examined using the NIS-Elements D 3.20 software tool.

### 3.12. Statistical Analysis

The antioxidant activity data for three independent measurements are reported as the mean ± standard deviation (SD). Analysis of variance (ANOVA) and Dunnett’s multiple-range tests were utilized to assess the statistical significance, and *p* < 0.05 was considered statistically significant. The immunoregulatory effects are expressed as the mean ± standard error (SE). SPSS 26.0 was used for statistical analysis, and *p* < 0.05 was defined as statistically significant.

## 4. Conclusions

In this study, RSM was successfully utilized to optimize the extraction parameters of LBPs. The optimum extraction conditions obtained were 100 °C extraction temperature, 75 min extraction duration, and 17 mL/g of water-to-wolfberry ratio. The experimental yield of the polysaccharides was 15.58 ± 0.19%, which was nearly identical to the predicted output (15.57%). Ultrafiltration membrane separation was used to obtain one oligosaccharide, LBPs-1 (1.912 kDa), and two polysaccharides, LBPs-2 (7.481 kDa) and LBPs-3 (46.239 kDa) from the crude polysaccharides.

The structural properties of LBPs-1, LBPs-2, and LBPs-3 were characterized by HPSEC-RI-MALS, HPAEC-PAD, FTIR, Congo red experiment, AFM, and NMR spectroscopy. Their chemical compositions, monosaccharide types, and FT-IR spectra were almost identical. However, the LBPs with different molecular weights possessed distinct chemical composition ratios, monosaccharide ratios, and triple-helix conformations. LBPs-1 and LBPs-2 were primarily composed of glucose, but LBPs-3 contained arabinose, galactose, and glucose. Additionally, the Congo red experiments revealed the presence of the triple-helix conformation only in LBPs-2 and LBPs-3, which was confirmed by AFM. Moreover, the NMR analysis revealed that the glycosyl residues of LBPs-2 contained →4)-β-Gal*p*-(1→, α-Glc*p*-(1→, →3)-α-Glc*p*-(1→, β-Glc*p*-(1→, →3,4)-β-Ara*p*-(1→, →3)-α-Ara*p*-(1→ and →4)-α-D-Glc*p*A-(1→, while the glycosyl residues of LBPs-3 contained →3,4)-α-Ara*p*-(1→, →3,4)-α-Gal*p*-(1→, →3)-α-Gal*p*-(1→, →4)-β-Ara*p*-(1→, β-Ara*p*-(1→, →3,4)-α-Gal*p*-(1→ and →3)-α-Glc*p*A-(1→.

In addition, in vitro antioxidant activities of LBPs were evaluated. The polysaccharides (LBPs-2 and LBPs-3), in this case, showed notably higher antioxidant capacities than the oligosaccharide (LBPs-1). LBPs-3 exhibited the highest ORAC and SOD activity, followed by LBPs-2 and LBPs-1. The superior ABTS^+^ radical scavenging activity of LBPs-2 could be attributed to the changes in the molecular weight, the chemical composition, the monosaccharide composition ratio, and the spatial structure. The immunomodulatory activity of LBPs-2 and LBPs-3 in the zebrafish model was investigated in vivo. LBPs-2 and LBPs-3 were able to enhance the immune function of the VTI-treated immunocompromised zebrafish by increasing the number of T-lymphocytes and macrophages. LBPs-2 and LBPs-3 are beneficial antioxidant and immunomodulatory agents for the pharmaceutical and functional food industries. These findings offer a new perspective on LBPs.

## Figures and Tables

**Figure 1 molecules-28-00701-f001:**
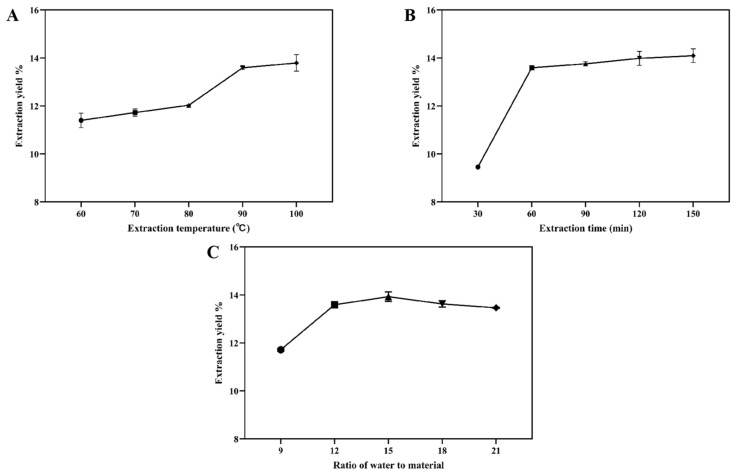
Effects of the extraction temperature (**A**), time (**B**), and water-to-raw material ratio (**C**) on the extraction yield of LBPs.

**Figure 2 molecules-28-00701-f002:**
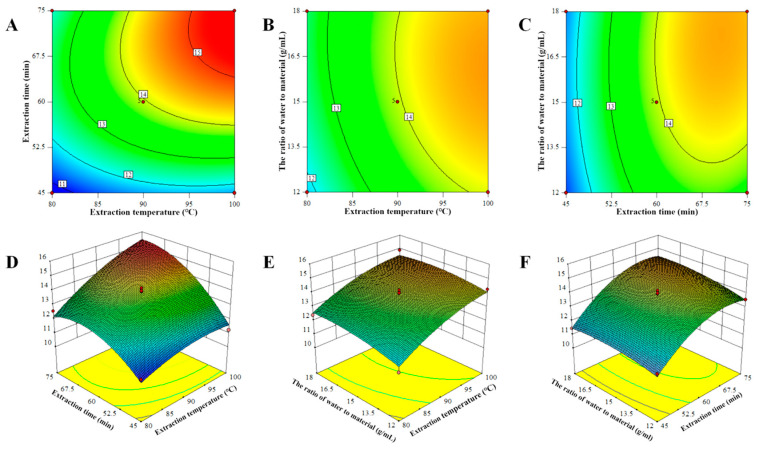
Three-dimensional response surface plots and two-dimensional contour plots showing the effect of interactions between variables on polysaccharide extraction yields. (**A**,**D**) display the influence of the extraction temperature and the time on the extraction yield of LBPs; (**B**,**E**) display the influence of the extraction temperature and the ratio of water to material on the extraction yield of LBPs; (**C**,**F**) display the influence of the time and the ratio of water to material on the extraction yield of LBPs.

**Figure 3 molecules-28-00701-f003:**
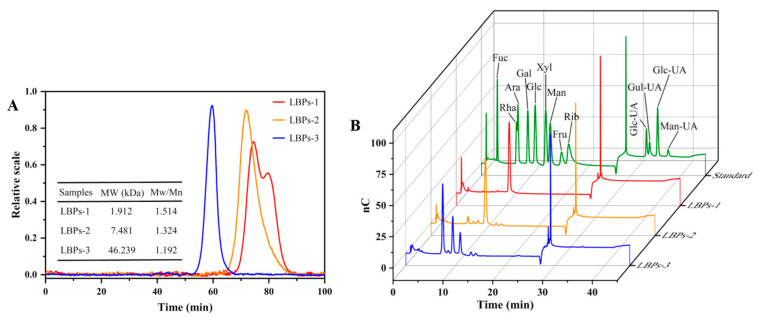
HPSEC-RID-MALS chromatogram of molecular weight (**A**) and HPAEC-PAD chromatograms of monosaccharides (**B**) of LBPs with three different molecular weights.

**Figure 4 molecules-28-00701-f004:**
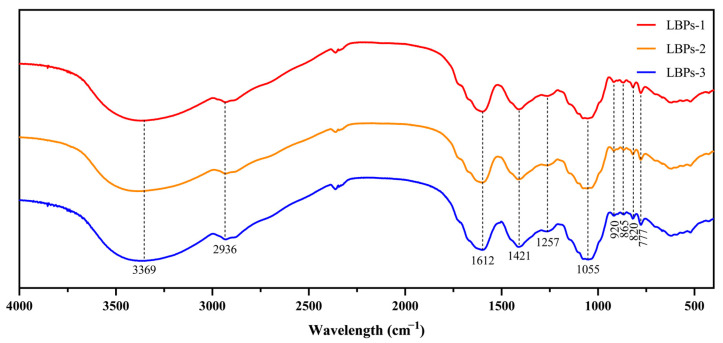
FT-IR spectra of LBPs with three different molecular weights.

**Figure 5 molecules-28-00701-f005:**
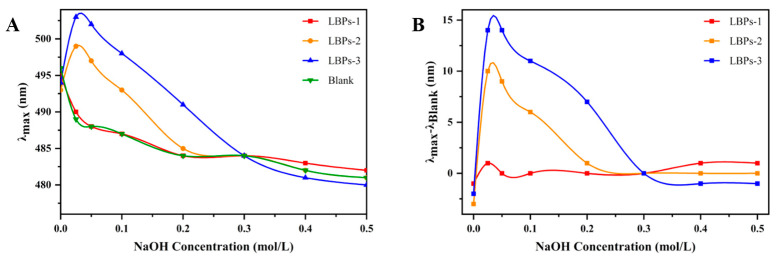
The maximum absorption wavelengths (**A**) and value changes versus the blank control (**B**) of the mixtures of Congo red and LBPs with three different weights at different concentrations of NaOH.

**Figure 6 molecules-28-00701-f006:**
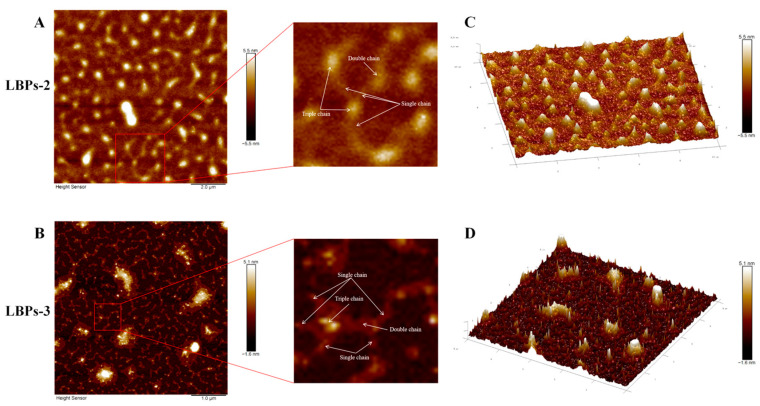
AFM planar images ((**A**,**B**): 10 μm), AFM 3D images ((**C**,**D**): 10 μm) of LBPs-2 and LBPs-3.

**Figure 7 molecules-28-00701-f007:**
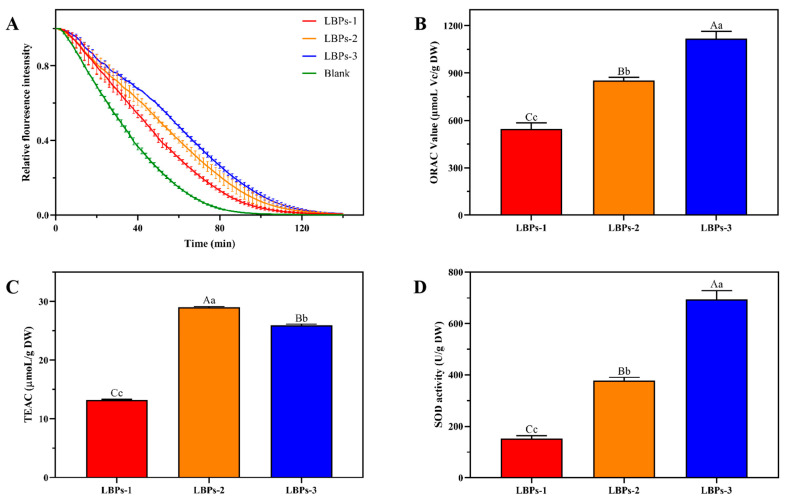
Relative fluorescence attenuation curve (**A**), ORAC value (**B**), scavenging effects on ABTS^+^ radicals (**C**), and SOD activity (**D**) of LBPs with three different molecular weights. Means within a column with different superscript upper- and lowercase letters differ significantly with *p* < 0.01 or *p* < 0.05, respectively.

**Figure 8 molecules-28-00701-f008:**
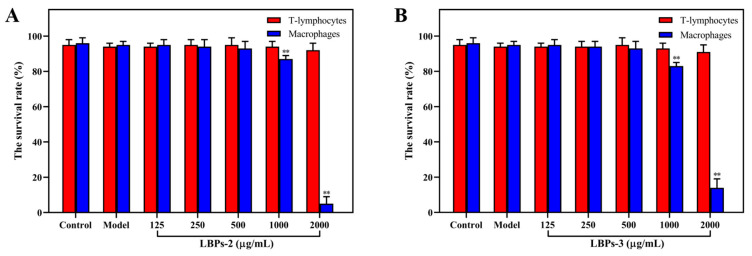
The survival rate of zebrafish treated by LBPs-2 (**A**) and LBPs-3 (**B**) with different concentrations. All data are presented as the mean ± SD (*n* = 30); ** indicate statistically significance differences at *p* < 0.01 compared to the control group.

**Figure 9 molecules-28-00701-f009:**
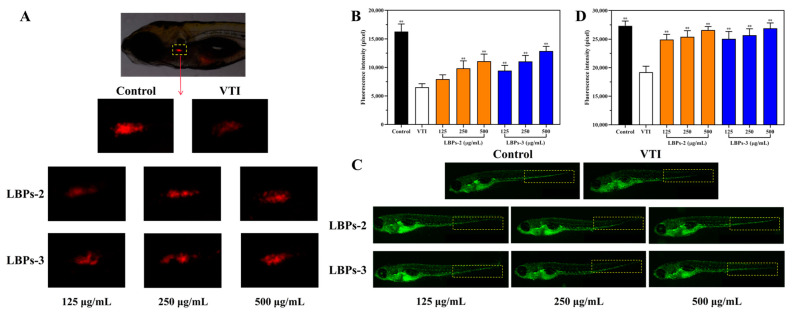
Effect of LBPs-2 and LBPs-3 on T-lymphocytes and macrophages in VTI-treated zebrafish. Fluorescence intensity of T-lymphocytes in zebrafish (**A**), Experimental results of the pro-proliferative effect of LBPs-2 and LBPs-3 on T-lymphocytes (**B**), Fluorescence intensity of macrophages in zebrafish (**C**), Experimental results of the pro-proliferative effect of LBPs-2 and LBPs-3 on T-lymphocytes (**D**). All data are presented as the mean ± SD (*n* = 10); ** indicate statistically significance differences at *p* < 0.01 compared to the VTI-treated group.

**Table 1 molecules-28-00701-t001:** Box–Behnken experimental design and results.

Number	Factor			Actual Value (%)	Predicted Value (%)
	A	B	C		
1	90 (0)	45 (−1)	12 (0)	11.35	11.18
2	80 (−1)	60 (0)	12 (0)	11.41	11.81
3	90 (0)	60 (0)	15 (0)	13.97	13.95
4	90 (0)	75 (1)	18 (1)	14.26	14.44
5	100 (1)	60 (0)	12 (−1)	14.26	14.07
6	90 (0)	60 (0)	15 (0)	13.61	13.95
7	90 (0)	60 (0)	15 (0)	13.98	13.95
8	90 (0)	60 (0)	15 (0)	14.20	13.95
9	100 (1)	75 (1)	15 (0)	15.24	15.44
10	90 (0)	60 (0)	15 (0)	14.00	13.95
11	90 (0)	45 (−1)	18 (1)	11.45	11.48
12	80 (−1)	75 (1)	15 (0)	12.57	12.20
13	90 (0)	75 (1)	12 (−1)	13.56	13.54
14	80 (−1)	45 (−1)	15 (0)	10.91	10.70
15	80 (−1)	60 (0)	18 (1)	12.40	12.59
16	100 (1)	60 (0)	18 (1)	14.88	14.49
17	100 (1)	45 (−1)	15 (0)	11.24	11.62

**Table 2 molecules-28-00701-t002:** ANOVA for response surface model.

Source	Sum of Squares	Degrees of Freedom	Mean Squares	F	*p*
Model	31.12	9	3.46	24.28	0.0002
A	8.70	1	8.70	61.12	0.0001
B	14.25	1	14.25	100.08	<0.0001
C	0.73	1	0.73	5.09	0.0586
AB	1.37	1	1.37	9.60	0.0173
AC	0.035	1	0.035	0.25	0.6354
BC	0.087	1	0.087	0.61	0.4593
A^2^	0.81	1	0.81	5.71	0.0482
B^2^	4.39	1	4.39	30.83	0.0009
C^2^	0.31	1	0.31	2.20	0.1815
Residual	1.00	7	0.14		
Lack of fit	0.81	3	0.27	5.94	0.0590
Pure error	0.18	4	0.046		
Cor. total	32.12	16			

**Table 3 molecules-28-00701-t003:** Effects of different molecular weights of LBPs on chemical properties and monosaccharide compositions.

Index	LBPs-1	LBPs-2	LBPs-3
Total sugar (%)	40.62 ± 1.48 ^Aa^	39.04 ± 1.26 ^Aa^	39.34 ± 1.55 ^Aa^
Protein (%)	13.31 ± 0.14 ^Cc^	22.23 ± 0.14 ^Bb^	24.19 ± 0.17 ^Aa^
Total polyphenol (%)	1.77 ± 0.05 ^Cc^	5.89 ± 0.23 ^Aa^	4.51 ± 0.15 ^Bb^
Monosaccharide composition (mol%)			
Fucose	-	0.38	0.50
Rhamnose	0.10	0.82	2.03
Arabinose	0.58	7.58	49.27
Galactose	0.31	3.93	28.35
Glucose	97.39	80.56	13.38
Xylose	0.25	2.49	2.42
Mannose	0.72	3.64	2.73
Glucuronic Acid	0.64	0.60	1.32

Note: Values are expressed as mean ± SD (*n* = 3). Means within a column with different superscript upper- and lowercase letters differ significantly with *p* < 0.01 and *p* < 0.05, respectively.

**Table 4 molecules-28-00701-t004:** Chemical shifts in resonances in the NMR spectra of LBPs-2.

Glycosyl Residues		Chemical Shifts (ppm)					
		1	2	3	4	5	6
→4)-β-Gal*p*-(1→	H	4.50	3.28	3.54	3.93	3.23	3.09/3.23
	C	102.92	71.23	74.02	78.17	74.96	63.77
α-Glc*p*-(1→	H	4.97	3.47	3.60	3.23	3.35	3.36
	C	103.13	69.36	71.12	69.77	73.00	62.84
→3)-α-Glc*p*-(1→	H	5.08	3.79	3.90	3.54	3.45	3.19
	C	101.41	76.09	81.79	71.06	69.34	53.83
β-Glc*p*-(1→	H	4.50	3.23	3.47	3.54	3.28	3.19
	C	102.49	74.02	69.36	70.63	72.35	53.83
→3,4)-β-Ara*p*-(1→	H	4.50	3.33	3.79	3.91	3.16	
	C	95.81	71.54	76.09	81.79	53.83	
→3)-α-Ara*p*-(1→	H	5.12	3.73	3.85	3.54	3.30	
	C	98.18	72.57	74.08	67.40	71.28	
→4)-α-D-Glc*p*A-(1→	H	5.08	4.06	3.71	4.20	4.06	-
	C	107.44	73.92	64.91	82.25	73.93	172.86

**Table 5 molecules-28-00701-t005:** Chemical shifts in resonances in the NMR spectra of LBPs-3.

Glycosyl Residues		Chemical Shifts (ppm)					
		1	2	3	4	5	6
→3,4)-α-Ara*p*-(1→	H	5.09	3.92	4.06	3.98	3.71	
	C	109.16/109.37	76.44	79.64	80.32	59.16	
→3,4)-α-Gal*p*-(1→	H	5.01	3.83	4.09	3.96	3.73	3.35
	C	107.44	76.66	83.76	80.75	76.50	71.92
→3)-α-Gal*p*-(1→	H	4.92	3.73	4.06	3.59	3.71	3.31
	C	107.22	74.94	81.18	77.30	74.94	70.85
→4)-β-Ara*p*-(1→	H	4.61	3.71	3.89	4.07	3.64/3.75	
	C	103.56	75.31	77.09	79.64	60.95	
β-Ara*p*-(1→	H	4.68	3.62	3.69	3.58	3.49	
	C	103.13	71.08	72.35	74.51	61.32	
→3,4)-α-Gal*p*-(1→	H	5.06	3.41	3.91	4.01	3.32	3.34
	C	102.49	75.63	81.18	83.98	74.33	68.15
→3)-α-Glc*p*A-(1→	H	5.19	3.72	4.62	3.61	4.49	-
	C	103.13	79.03	80.10	77.52	82.04	171.79

## Data Availability

The data presented in this study are available on request from the corresponding author.

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
