# Peer review of "Comparative Study on the Structural Properties and Bioactivities of Three Different Molecular Weights of Lycium barbarum Polysaccharides"

_molecules, 2023, doi:10.3390/molecules28020701_

Round 1
Reviewer 1 Report
The main question addressed by the research is the characterization, both from physic-chemical and biological point of view, of three polysaccharides extracted from Lycium barbarum. In particular, the effect of different molecular weights was considered.
I consider that this topic is relevant in this field, since the physic-chemical characterization is also of high value in order to better understand the biological activities.
Compared to other published material, the AFM investigation is added. The conclusions are consistent with the data presented. References are appropriate .
The manuscipt is of scientific interest. Minor suggestions:
Line 45: were dependent
Line 245-248: check the repeating unit. It seems that some linkage positions are not expressed.
Author Response
The main question addressed by the research is the characterization, both from physic-chemical and biological point of view, of three polysaccharides extracted from Lycium barbarum. In particular, the effect of different molecular weights was considered.
I consider that this topic is relevant in this field, since the physic-chemical characterization is also of high value in order to better understand the biological activities.
Compared to other published material, the AFM investigation is added. The conclusions are consistent with the data presented. References are appropriate.
The manuscript is of scientific interest. Minor suggestions:
Line 45: were dependent
Response: We highly appreciate the reviewer’s positive feedback and point out the mistake. We have modified the word and highlighted with yellow.
Line 245-248: check the repeating unit. It seems that some linkage positions are not expressed.
Response: Thank reviewer for the very careful consideration. We have carefully checked the repeating units. There occurred “α-Glcp-(1→” in line 245, “β-Glcp-(1→” in line 246, “β-Arap-(1→” in line 248 in the paper. We inferred that these glucoses and arabinoses linked at the terminal of LBPs-2 or LBPs-3. As shown in Table 5, the chemical shift of carbons in these glycosidic residues did not transfer to downfield except the anomeric carbons. In the HMBC spectra (Fig. 7E and 8E), there only observed the long-range correlations of the anomeric protons or carbons. Consequently, we reported the linkage positions of these repeating units of polysaccharides as shown in the manuscript.
Reviewer 2 Report
The manuscript “Comparative study on the structural properties and bioactivities of three different molecular weights of Lycium barbarum polysaccharides” has higher reference value. But the manuscript has some problems that need to be clarified. Some minor points also need to be considered as detailed below.
1. The reviewer thinks that it is difficult to compare the correlation between the structural characteristics and the activity of the three components obtained only by the same extraction process. Is it possible to consider using different processes or modifying methods to obtain more polysaccharides with different molecular weights, so as to make the comparison easier?
2. The separation method is the main factor affecting the molecular weight of polysaccharides, especially the polysaccharides obtained by ultrafiltration, which may be composed of a series of polysaccharides with different molecular weight distribution, and generally should be further purified and confirmed by gel chromatography.
3. Line 20 ʻThe purpose of the study was to determine how different molecular weights affected the structure and bioactivity of the Lycium barbarum polysaccharides (LBPs)ʼ. Is this the purpose of this study? If so, what does the author hope to achieve in lines 75-159?, At the same time, how does the molecular weight affect the chemical composition and monosaccharide composition of polysaccharides? Glycosidic bond and configuration?
4. Line 170-172 Percentages of LBPs-1, LBPs-2, and LBPs-3 should be given.
5. Line 187-188 It should be Figure 3B. And please check the peak symbols of the standards in Figure 3B.
6. Line 192-194 ʻThe results suggested that the fraction of glucose in polysaccharides decreased with increasing molecular weight, whereas the proportions of rhamnose, arabinose, and galactose increased.ʼ Is the correlation between monosaccharide composition and molecular weight obtained from two polysaccharide components too far-fetched?
Author Response
The manuscript “Comparative study on the structural properties and bioactivities of three different molecular weights of Lycium barbarum polysaccharides” has higher reference value. But the manuscript has some problems that need to be clarified. Some minor points also need to be considered as detailed below.
- The reviewer thinks that it is difficult to compare the correlation between the structural characteristics and the activity of the three components obtained only by the same extraction process. Is it possible to consider using different processes or modifying methods to obtain more polysaccharides with different molecular weights, so as to make the comparison easier?
Response 1: We would like thank the reviewer’s positive feedback and the constructive suggestion. We agree that it need more LBPs with different Mw to clarify the relationship between the molecular weight and the bioactivity. In the present study, 1 oligosaccharide (LBPs-1) and 2 polysaccharides (LBPs-2 and LBPs-3) were separated from Lycium barbarum extract according to the patent technique developed by our group. LBPs-2 and LBPs-3 exhibited higher antioxidant capacities than LBPs-1. The observation indicated that the molecular weight would be one of factors influenced the bioactivities of LBPs. Although this study is an individual case, it is interesting that LBPs with different Mw possessed different monosaccharides composition. In the future research, we will try to obtain more polysaccharides with different Mw using different processes. Consequently, this constructive comment of reviewer is of great guiding significance to our research.
- The separation method is the main factor affecting the molecular weight of polysaccharides, especially the polysaccharides obtained by ultrafiltration, which may be composed of a series of polysaccharides with different molecular weight distribution, and generally should be further purified and confirmed by gel chromatography.
Response 2: We are grateful to the reviewer for pointing out this issue. The column chromatography is the most common technique to purify polysaccharides, although it is always time-consuming. The ultrafiltration is more rapid, more reproducible, and less labor intensive. Consequently, we proposed to purify the LBPs with the ultrafiltration. Prior to purification, we determined the molecular weight distribution of the crude LBPs, which allowed to selected the suitable ultrafiltration tubes for LBPs’ separation. The polymer’s purity is always represented with the polydispersity. The polydispersity of LBPs-2 and LBPs-3 were confirmed as 1.324 and 1.192 by HPSEC-RID-MALLS, respectively. The results indicate that it is possible to purify polysaccharide by ultrafiltration.
- Line 20 ʻThe purpose of the study was to determine how different molecular weights affected the structure and bioactivity of the Lycium barbarum polysaccharides (LBPs)ʼ. Is this the purpose of this study? If so, what does the author hope to achieve in lines 75-159?, At the same time, how does the molecular weight affect the chemical composition and monosaccharide composition of polysaccharides? Glycosidic bond and configuration?
Response 3: We are grateful to the reviewer for pointing out this issue. The nature of LBPs can vary depending on the extraction procedure. The preparation method reported in line 75-159 provided an extraction procedure in detail and allowed researchers to repeat the experiments. Meanwhile, the product obtaining with this preparation can provide a consistent sample for LBPs separation.
We are sorry for the inaccurate description in line 20. The molecular weight, monosaccharides composition, glycosidic bond and configuration are the nature of LBPs and not logically related. Due to the complexity of the polysaccharides’ structures, it is a challenging to determine which structural characteristics may influence bioactivity. Consequently, in our opinion, it should characterize the structural features as thoroughly as possible while describing the biological effects. In the present study, we tried our best to elucidate the chemical structures of LBPs-2 and LBPs-3 in detail. We objectively reported an interesting observation that LBPs with different molecular weight possessed different monosaccharides composition. In the revised manuscript, we have modified the description and highlighted with yellow.
- Line 170-172 Percentages of LBPs-1, LBPs-2, and LBPs-3 should be given.
Response 4: Thank reviewer for this good suggestion. The percentages of LBPs-1, LBPs-2 and LBPs-3 are about 42.62%, 3.41% and 10.22% in the crude LBPs, respectively. According to the suggestion, we have added this data in the revised manuscript and highlighted with yellow.
- Line 187-188 It should be Figure 3B. And please check the peak symbols of the standards in Figure 3B.
Response 5: Thank the reviewer for the careful review. We are very sorry for our careless writing. We have corrected the mistakes in the revised manuscript and highlighted with yellow.
- Line 192-194 ʻThe results suggested that the fraction of glucose in polysaccharides decreased with increasing molecular weight, whereas the proportions of rhamnose, arabinose, and galactose increased.ʼ Is the correlation between monosaccharide composition and molecular weight obtained from two polysaccharide components too far-fetched?
Response 6: We are grateful to the reviewer for pointing out this issue. We also agree that the inference is far-fetched. We are sorry for the careless description. The inference has been modified as “ The results suggested that, in this case, the proportion of glucose was dominant in polysaccharides with low Mw, and replaced by arabinose and galactose in high Mw LBPs.” The correction was highlighted with yellow in the revised manuscript.
Reviewer 3 Report
This work carried out a Comparative study on the structural properties and bioactivities of three different molecular weights of Lycium barbarum polysaccharides. This is an interesting and systematic work. Here are some improvements that need to be considered:
1. There are many study on Lycium barbarum polysaccharides, what is the novelty and improvement of this work, the authors should state clearly in introduction and conclusion.
2. The relationship between bioactivities and molecular weights of Lycium barbarum polysaccharides was not clear, the authors should stated them clearly to improve the impact of the this work.
3. There are many errors in the manuscript, the figures are chaotic. In fact, figure 1 should be graphic abstract. The figures should be put together in manuscript. The tables should also be put in the manuscript.
4. The whole manuscript should be improved.
Author Response
This work carried out a Comparative study on the structural properties and bioactivities of three different molecular weights of Lycium barbarum polysaccharides. This is an interesting and systematic work. Here are some improvements that need to be considered:
- There are many study on Lycium barbarum polysaccharides, what is the novelty and improvement of this work, the authors should state clearly in introduction and conclusion.
Response 1: We would like to thank reviewer for the positive feedback and the constructive recommendation. Due to the complexity of the polysaccharides’ structures, it is a challenging to determine which structural characteristics may influence bioactivity. Meanwhile, the nature of LBPs can vary depending on the extraction procedure. Consequently, it should describe the preparation method and characterize the structural features as thoroughly as possible while describing the biological effects. In the present study, we systematic described the obtaining procedure of LBPs with different molecular weight and elucidate the chemical structures of LBPs-2 and LBPs-3 in detail, while the bioactivity assays were conducted. This would provide valuable information to understand the molecular mechanism of LBPs. That is the aim and improvement of the study.
- The relationship between bioactivities and molecular weights of Lycium barbarum polysaccharides was not clear, the authors should state them clearly to improve the impact of the this work.
Response 2: Thank the reviewer for the constructive recommendation. According to our findings, the molecular weight is not the single factor influencing the bioactivities of LBPs. The polysaccharides (LBPs-2 and LBPs-3), in this case, really showed higher antioxidant capacities than those of oligosaccharide (LBPs-1). According to the suggestion, we added this inference in Abstract and Conclusion and highlighted with yellow in the revised manuscript.
- There are many errors in the manuscript, the figures are chaotic. In fact, figure 1 should be graphic abstract. The figures should be put together in manuscript. The tables should also be put in the manuscript.
Response 3: Thank reviewer for these very careful considerations. We also apologize for our careless writing. The mistakes have been modified after this revision.
- The whole manuscript should be improved.
Response 4: Thank reviewer for these very careful considerations. The revised manuscript has been undergone extensive English revisions by the editing services of MDPI.
Round 2
Reviewer 1 Report
Figures regarding NMR are not very clear.
1H spectrum displays many peaks that are not clearly readable. I thinkm this is not useful. Similar considerations for 2D spectra.
Reviewer 3 Report
The manuscript has been improved and could be accepted in current form.